# Reliability and Construct Validity of the Yale Pharyngeal Residue Severity Rating Scale: Performance on Videos and Effect of Bolus Consistency

**DOI:** 10.3390/diagnostics12081897

**Published:** 2022-08-04

**Authors:** Sara Rocca, Nicole Pizzorni, Nadia Valenza, Luca Negri, Antonio Schindler

**Affiliations:** 1Department of Biomedical and Clinical Sciences, Università degli Studi di Milano, 20157 Milan, Italy; 2Department of Pathophysiology and Transplantation, Università degli Studi di Milano, 20122 Milan, Italy

**Keywords:** deglutition disorders, fiberoptic endoscopic evaluation of swallowing, reliability

## Abstract

The Yale Pharyngeal Residue Severity Rating Scale (YPRSRS) provides an image-based assessment of pharyngeal residue in the fiberoptic endoscopic evaluation of swallowing (FEES). Its performance was investigated only in FEES frames. This study analyzed the reliability and construct validity of the YPRSRS in FEES videos and the influence of bolus consistency. Thirty pairs of FEES videos and frames, 8 thin liquids (<50 mPa·s), 11 pureed (2583.3 mPa·s at 50 s^−1^, 697.87 mPa·s at 300 s^−1^), and 11 solid food; were assessed by 29 clinicians using the YPRSRS; 14 raters re-assessed materials at least 15 days from the first evaluation. Construct validity and intra-rater reliability were assessed using weighted Cohen’s Kappa. Inter-rater reliability was assessed using weighted Fleiss Kappa. Construct validity and inter-rater reliability were almost perfect or excellent for frames (0.82 ≤ k ≤ 0.89) and substantial or intermediate to good for videos (0.67 ≤ k ≤ 0.79). Intra-rater reliability was almost perfect for both frames and videos (k ≥ 0.84). Concerning bolus consistency, thin liquids had significantly lower values of construct validity, intra-, and inter-rater reliability than pureed and solid food. Construct validity and inter-rater reliability were significantly lower for solid food than for pureed food. The YPRSRS showed satisfactory reliability and construct validity also in FEES videos. Reliability was significantly influenced by bolus consistency.

## 1. Introduction

Pharyngeal residues and penetration/aspiration are the two most important signs of swallowing disorder; aspiration and pharyngeal residue are highly correlated [1]. Pharyngeal residues can be caused by various factors, such as: upper esophageal sphincter dysfunction, inadequate tongue base retraction, and impaired pharyngeal bolus propulsion [2]. Fiberoptic endoscopic examination of swallowing (FEES) is considered, together with the videofluoroscopic examination of swallowing (VFSS), the “gold standard” for the diagnosis of dysphagia [3]. FEES allows directly observing the pharyngeal phase of oropharyngeal swallow to identify possible signs of dysphagia, such as: penetration, aspiration, and pharyngeal residues [4]. FEES showed greater sensitivity than VFSS regarding aspiration, penetration, and residue evaluation [5,6]. The endoscopic view allows recognizing the residue sites within the pharynx and larynx, identifying when the risk for aspiration is higher [5]. The interpretation of signs of dysphagia detected during FEES or VFSS is often subjective. Thus, different rating scales, typically visuoperceptual measures, have been introduced to provide a common language among clinicians and enable a reliable assessment. Usually, temporal, spatial, and volumetric variables are employed [6,7]. Available scales to evaluate the pharyngeal residues severity in FEES include ordinal scales [8,9,10,11,12], estimation scales [13,14] and binary scales [15]. Recently, in two reviews focused on psychometric qualities of visuoperceptual scales, the Yale Pharyngeal Residue Severity Rating Scale (YPRSRS) [11] showed good/excellent reliability [7] and met the criteria for a valid and reliable residue severity rating scale based on FEES [16]. The YPRSRS is an image-based system to assess the amount of residue in the valleculae and pyriform sinus. The authors consider the scale generalizable and applicable to all age groups thanks to the operational definitions, which refer to the residue and anatomical indices. To date, the YPRSRS has been validated in English [11], German [17], and Turkish [18]. The German and Turkish studies were based on FEES images from the original validation study, which displayed bolus residue of yellow pudding, white milk, or no residue. However, previous studies did not provide information on the degree of influence bolus consistency can have on the evaluation of the residue, nor on the application of YPRSRS directly on FEES videos. These insights are important because they reflect an authentic clinical assessment, often in real-time or recorded FEES videos—including different consistencies. In FEES videos assessment, clinicians must select at what time to assess pharyngeal residues and their severity; the temporal component of the videos, which is absent in the frames, could make them more complex to evaluate. Furthermore, the literature suggests that the bolus’s consistency can impact the raters’ agreement when using visuoperceptual scales [19]. Additionally, rheological properties of foods, such as viscosity, are known to influence the risk of aspiration and the frequency of post-swallow pharyngeal residues [20]. Bolus modification is one the most recommended strategies for dysphagia management. Thus, it is relevant for clinicians to be able to reliably assess residue when testing different consistencies. Regarding the definition of consistency, the International Dysphagia Diet Standardisation Initiative (IDDSI) framework [21] provides a common terminology for liquid and food consistencies based on qualitative definitions of food texture. Additionally, viscosity can be objectively measured with a viscometer and expressed according to the International System (IS) of Units in Pascal-second (Pa s), (N s)/m^2^ or kg/(m s) [20].

This study aimed to investigate the psychometric properties of the Yale Pharyngeal Residue Severity Rating Scale (YPRSRS) based on FEES frames and respective videos. In particular, both the reliability and the construct validity of the YPRSRS were analyzed and compared: (i) between frames and videos; and (ii) among different bolus consistencies. The following hypotheses were formulated: (i) the YPRSRS can be a valid and reliable scale to assess residue in FEES videos, although videos may be more challenging to rate than frames; (ii) validity and reliability are influenced by bolus consistency. Regarding the study’s clinical implications, the validation in videos and different bolus consistencies could be helpful for clinicians in their clinical practice to promote a complete and replicable assessment of pharyngeal residues.

## 2. Materials and Methods

This project was carried out following the Declaration of Helsinki of the World Medical Association (WHO). Consent of the Ethics Committee of the University of Milan (protocol code 102/2, date of approval 17 November 2020) was obtained. All data were processed in a pseudonymized form by assigning an alphanumeric code to each rater.

### 2.1. Yale Pharyngeal Residue Severity Rating Scale (YPRSRS)

The YPRSRS is an ordinal scale to identify the location and rate the severity of pharyngeal residues observed through FEES in the post-swallowing phase [11]. The scale comprises two scores for pharyngeal residues in the valleculae and pharyngeal residues in the piriform sinus. The severity definitions are distributed on a 5-point scale (none, trace, mild, moderate, severe). For each level, an operational description, an anchor image, and a percentage of residue are provided, both for the valleculae and the piriform sinuses (Table 1).

### 2.2. Frames and Videos Selections

Employed videos and images were selected from the department’s archival material. The FEES was performed by a Phoniatrician using a flexible transnasal endoscope Olympus XION EF-N flexible fiberscope (XION GmbH, Berlin, Germany) attached to an EndoSTROBE camera (XION GmbH, Berlin, Germany) and recorded as an AVI format anonymously.

The evaluations were carried out with thin liquids (5–10–20 mL of blue-dyed water room-temperature × 3 trials for each volume; IDDSI 0; <50 mPa·s at 50 s^−1^ and 300 s^−1^), pureed food (5–10–20 mL of Crème Line Valilla Nutrisens—Nutrisens Italia SRL, Turin, Italy—pudding × 3 trials for each volume; IDDSI 4; 2583.3 ± 10.41 mPa·s at 50 s^−1^ and 697.87 ± 7.84 mPa·s at 300 s^−1^), and regular food (half 8 g of Frollini Monviso—Monviso group SRL, Andezeno, TO, Italy—biscuit × 2 trials; IDDSI 7 Regular). The viscosity analyses were performed with the Haake Viscotester 550 (Thermo Electron GmbH, Dieselstr, Germany); viscosities below 300 mPa·s were performed with the system MV1 (gap: 0.96 mm) and viscosities over 300 mPa·s with the system SV1 (gap: 1.45 mm). The shear rate for the swallowing process can range from 1 to 1000 s^−1^ [22]. In this work, as according to previous studies [23,24], the values of 50 and 300 s^−1^ were used to reflect viscosity either at the oral stage or the pharyngeal stage of swallowing. From the FEES recordings, only the video clips that recorded the swallowing acts of the 5 mL for thin liquids and pureed food were selected for the validation study. For frame selection, a post-swallow frame was selected from each video at the end of the last visible swallow. In assessing videos, raters were asked to assign a YPRSRS score to the pharyngeal residue observed at the end of the swallowing series. Two experts, a phoniatrician and a speech and language pathologist (SLP), with at least ten years of experience in dysphagia, independently assessed a total of 70 pairs of videos and frames by assigning to each of them a level of the YPRSRS. The judgment of two experts was used as the gold standard for the construct validity analysis, according to the procedure used in the YPRSRS validation in the German paper [17]. Only the frames and videos that had obtained a perfect agreement on the score from the experts were selected. A total of 44 frames and video pairs were attributed the same YPRSRS score by the two experts. Thirty frames and videos pairs were selected and used for the validation purpose according to the following criteria: (i) 15 valleculae and 15 pyriform sinuses, (ii) 3 pairs for each YPRSRS level for both valleculae and pyriform sinus, and (iii) 8 pairs with thin liquid (IDDSI 0), 11 pairs using pureed food (IDDSI 4), and 11 pairs with solid food (IDDSI 7).

### 2.3. Raters

Raters were recruited among clinicians from different institutions. The inclusion criteria consisted of professional activity as either an SLP, otolaryngologist, phoniatrician, or resident otolaryngologist with a minimum clinical experience of 1 year in dysphagia. Data on the number of years of experience, the frequency with which raters perform or attend FEES exams, any post-basic training courses on dysphagia, and previous clinical experience with the YPRSRS scale were collected.

### 2.4. Procedure

Ratings were obtained via a google form and frames and videos were in random order. At recruitment, each rater was assigned an alphanumeric code. 50% of the raters (n = 15) were randomly selected to assess videos and frames twice; with at least 15 days between the first and the second evaluation. At each assessment, the raters were asked to indicate their alphanumeric code for identification in each form. All forms contained guidance for the assessment, the scale, and the anchor images. Raters were asked to view the images and videos in full-screen mode. The forms for the second evaluation were sent to the participants two weeks after the completion of the first evaluation

### 2.5. Statistical Analysis

The analyses were carried out using the IBM SPSS v26.0 ^®^ software for Windows (SPSS Inc., Chicago, IL, USA) and R software v.4.2.0 [25]. Construct validity, intra-rater reliability, and inter-rater reliability were calculated for videos, frames, and different consistencies, and raters background (SLP versus Medical Doctor, MD).

Construct validity was calculated with Cohen’s Kappa weighted (quadratic weighting) [26] by analyzing the agreement between each rater (first evaluation) and the experts.

The intra-rater reliability was calculated with the weighted Cohen’s Kappa (quadratic weighting) for the 15 raters who assessed the FEES materials twice.

Average Cohen’s Kappa was calculated from the single raters’ Cohen’s Kappa. The distribution of the Cohen’s Kappa was compared between videos and frames using the *t*-test and among bolus consistencies using the one-way analysis of variance (ANOVA) with posthoc Tukey HSD. Significance was set at *p* < 0.05.

The inter-rater reliability was determined by Fleiss Kappa [27] with quadratic weighting. As a first step, the level of agreement among all raters in assessing the amount of residue in the valleculae and the pyriform sinus was calculated irrespectively of bolus consistencies. This procedure was repeated both for frames and video evaluations. Subsequently, for valleculae and pyriform sinus, Fleiss Kappa indices associated with frames and video evaluations were compared using paired sample *t*-tests based on the linearization method for correlated agreement coefficients [28].

Raters’ level of agreement was calculated for thin liquid, pureed, and solid food in frames and video evaluations separately to test the influence of bolus consistency on inter-rater reliability. Subsequently, a one-way analysis of variance (ANOVA) was employed to compare Fleiss Kappa values; Tukey’s HSD method was implemented to correct the significance level for posthoc pairwise comparisons.

Significant differences in validity and reliability according to raters’ backgrounds were inspected by means of independent *t*-tests.

Concerning Cohen’s Kappa statistics, the levels of agreement were determined according to the following criteria: Kappa values of 0 were considered to indicate poor agreement, 0.00–0.20 slight agreement, 0.21–0.40 fair agreement, 0.41–0.60 moderate agreement, 0.61–0.80 substantial agreement, 0.81–1.00 almost perfect agreement [26]. As for the Fleiss Kappa, the following benchmark was adopted: <0.40 poor, 0.40–0.75 intermediate to good, >0.75 excellent [27].

## 3. Results

### 3.1. Raters’ Characteristics

A total of 29 clinicians participated in this study as raters. 20 were SLPs (29.74 ± SD 5.84 years; 95% female), 5 were otolaryngologists (39.60 ± SD 4.67 years, 40% female), and 4 were resident medical doctors in otolaryngology (30.54 ± SD 4.18 years, 50% female). Data on the characteristics of the participants are collected in Table 2.

### 3.2. Reliability and Validity in Videos and Frames

Results concerning reliability and validity in videos and frames are reported in Table 3, Table 4 and Table 5.

The construct validity results showed almost perfect agreement for frames (k > 0.81) and substantial agreement for videos (k > 0.61) for both locations (Table 3). Frames’ values were significantly higher than video values for valleculae and pyriform sinuses.

Kappa values of intra-rater reliability of raters regarding frames and videos were almost perfect (k > 0.81). No significant differences were found between frames and videos (Table 4).

As for inter-rater reliability among all professional raters at the first rating, it was excellent for frames in both valleculae and pyriform sinus locations; the agreement was intermediate to good for videos (Table 5). When Fleiss Kappa indices associated with frames and videos were compared (Table 5), no significant difference was found for valleculae; as for pyriform sinus, raters’ level of agreement for frames was significantly higher than for videos (although the *p* level was close to the 0.05 significance threshold).

### 3.3. Validity and Reliability According to Raters’ Background

The results based on raters’ background are reported in Table 6, Table 7 and Table 8.

The construct validity Kappa statistics in SLP and MD ranged from substantial agreement to almost perfect agreement. No statistically significant differences were observed between the two groups (Table 6).

The values of intra-rater reliability ranged from 0.87 to 0.92 for SLP and from 0.74 to 0.92 for MD. The *t*-test showed a significant difference in pyriform sinus frames rating between the two groups (Table 7).

Additionally, both groups’ Kappa values of inter-rater reliability ranged from intermediate to good to excellent. *t*-test results did not show significant differences between the groups (Table 8).

### 3.4. Influence of Bolus Consistency

The influence of bolus consistency on construct validity, intra-rater reliability, and inter-rater reliability was also analyzed; results are reported in Table 9, Table 10 and Table 11, respectively.

Concerning construct validity, the agreement with the experts in frames ranged from 0.56 to 0.88; a significant omnibus difference among agreement indices was found (F (1.70) = 46.43; *p* < 0.001). Videos values ranged from 0.44 to 0.88, with an ANOVA omnibus test result showing a significant difference (F (2.94) = 20.74; *p* < 0.001). Post-hoc analyses showed significantly lower values with thin liquids (IDDSI 0), both in videos and in frames, when compared with pureed food (IDDSI 4) and solid food (IDDSI 7). Moreover, solid food videos (IDDSI 7) had significantly worse values than pureed foods (IDDSI 4) (Table 9).

Intra-rater reliability values ranged from 0.53 to 0.89 for frames and from 0.46 to 0.89 for videos, with a significant omnibus difference among Cohen indices observed for frame (F (1.10) = 14.60; *p* < 0.001), and videos (F (1.55) = 15.22; *p* < 0.001). The posthoc test showed thin liquids’ (IDDSI 0) values for videos and frames significantly lower than other consistencies (Table 10).

As for inter-rater reliability, agreement among all raters was evaluated for thin liquid, pureed, and solid food separately (Table 11). Concerning frame evaluations, Fleiss Kappa indices ranged from 0.38 to 0.84, with an ANOVA omnibus test result showing a significant difference among agreements (F (2.27) = 10.27; *p* < 0.001). Results from posthoc analyses showed a significant lower agreement value for thin liquid (IDDSI 0) when compared with both pureed (IDDSI 4); and solid food (IDDSI 7); however, no difference was observed between pureed (IDDSI 4) and solid food (IDDSI 7).

Concerning video evaluations, inter-rater reliability indices ranged from 0.22 to 0.81. Similarly to the frames’ results, a significant omnibus difference among Fleiss indices was observed (F (2.27) = 9.28; *p* < 0.001). When agreement coefficients were compared pairwise, pureed food (IDDSI 4) showed higher Fleiss Kappa values than both thin liquid (IDDSI 0) and solid food (IDDSI 7). No significant differences were detected between thin liquid (IDDSI 0) and solid food (IDDSI 7) agreement indices.

## 4. Discussion

For the first time, psychometric characteristics of the YPRSRS were tested on FEES videos and different bolus consistencies. The scale showed substantial to almost perfect construct validity and reliability in both videos and frames. These results confirm the initial hypothesis that it is possible to use the YPRSRS for reliably rating pharyngeal residues in videos. Regarding the effect of bolus consistencies, thin liquids (IDDSI 0) had lower construct validity and reliability values than the other consistencies; additionally, solid food had significantly lower construct validity and inter-rater reliability values than pureed food only for video evaluations.

Concerning the psychometric properties of the YPRSRS on videos, there was a trend of lower levels of reliability and construct validity of videos than frames, which reached statistical significance for construct validity for both locations and inter-rater reliability for pyriform sinus. Lower reliability values could be caused by the greater complexity of videos than frames. Indeed, videos contain more information to be processed and interpreted than static images. They also require several skills in recognizing the various phases and swallowing timing. Moreover, a video-based assessment is more representative of actual clinical practice. When assessing pharyngeal residue in FEES videos, the timing of residue assessment needs to be correctly identified. In the present study, raters were asked to assess the residue at the end of the swallowing series. However, different reliability and construct validity values could have been obtained in case residues were assessed after the first swallow. Future studies should analyze the impact of the different timing of residue assessment on the reliability of the YPRSRS. The project did not provide the use of anchors for the videos. It would be interesting, in the future, to develop short anchor videos and verify if having these can increase the reliability of the scale in the videos.

Construct validity and reliability reached almost perfect values in the frames. These values are comparable to those of previous studies. Neubauer et al. [11] reported a construct validity of 0.951 in the valleculae frames (the value in this study is 0.89) and of 0.908 in the piriform sinuses (in this study, 0.87), and Gerschke [17] found almost perfect Kappa values for both sites (k > 0.90). For intra-rater reliability Neubauer et al. [11] reported Kappa values of 0.957 for vallecula (in this study 0.93) and 0.854 for pyriform sinus (in this study 0.84), German values were of 0.963 for valleculae and 0.944 for pyriform sinus [17]. Lastly, Neubauer et al. [11] found inter-rater reliability values of 0.868 and 0.751 for valleculae and pyriform sinus (in this study, 0.85 and 0.82), German values were 0.928 and 0.938 for valleculae and pyriform sinus [17].

The “best-of-the-best” criterion of previous studies [17], for which the images with the best quality are chosen and those with lower quality are excluded, was not followed in this work’s choice of videos and images. The quality of some frames and videos was not optimal, making the rating more complex but more likely similar to and consistent with everyday clinical practice. However, from the results, the reliability seems adequate even with images of less than perfect quality.

Analyses among groups with different backgrounds revealed Kappa values that were, for all except one comparison, not statistically different among SLPs and MDs. These results suggest that the scale can be used reliably by professionals with different backgrounds. As in Italy only MDs can perform FEES, the high values of construct validity and reliability reached for SLPs suggests that, regardless of the professional that performs FEES in clinical practice, the assessment of residues using the YPRSRS is a relatively easy task for SLPs working with patients with dysphagia. A significant difference emerged between SLPs and MDs only for the intra-rater reliability values of pyriform sinus frames, with higher reliability values for the SLPs. However, the considerable difference in the sample size between the two groups is worth considering and further studies with balanced groups should confirm our results. These results are consistent with those reported in a previous study [29] in which the reliability of SLPs and radiologists (RADs) to identify different signs of dysphagia and the presence of dysphagia in VFSS samples, before and after a training, was compared; no significant differences were recorded between SLPs and RADs in pre-training analyses.

Bolus consistencies influenced YPRSRS psychometric properties. Thin liquids (IDDSI 0) showed significantly lower reliability and construct validity than other consistencies. Due to their rheology, thin liquids could result in a more difficult residue assessment [19]. In this study, blue-dyed water was used, using different types of liquid, such as milk and barium, which could give different results [30].

Moreover, solid food values (IDDSI 7) in inter-rater reliability were significantly lower than pureed food values (IDDSI 4). Solid food boluses (IDDSI 7) are often divided into multiple swallows and may require additional clearing swallows; this, in videos, may make it more challenging for raters to choose the right time to score the residue with solid food compared to pureed food. In general, the best values were found for boluses of pureed food (IDDSI 4) in videos and frames. It should be noted that the anchor images of the original study were used, and only those with pureed food (IDDSI 4) were represented. Thus, the raters did not have reference examples on which to rely for the evaluation of thin liquids (IDDSI 0) and solid food (IDDSI 7), which could have made the residue assessment more challenging for these consistencies. 

This study has some limitations that need to be mentioned. The results were not analyzed considering the influence of years of experience. In Italy, the FEES procedure is a medical act; therefore, the limited number of medical doctors compared to the number of SLPs can be considered a study limitation. As previously mentioned, no anchor images were developed for thin liquids and solid foods and anchor videos were lacking. Lastly, for some frames and videos, the image quality was not optimal, which can be assumed to have affected reliability.

## 5. Conclusions

The YPRSRS can be reliably used to assess the severity of pharyngeal residue both in FEES frames and videos. In addition, clinicians should be particularly meticulous when evaluating thin liquid residues, for which it may be more challenging to assign a score reliably. Overall, it is possible to consider the results of this study as encouraging and positive for expanding clinicians’ skills in the field of dysphagia and providing them with adequate tools.

## Figures and Tables

**Table 1 diagnostics-12-01897-t001:** Severity definitions for valleculae and pyriform sinus residues.

Valleculae			
**I**	None	0%	No residue
**II**	Trace	1–5%	Trace coating of the mucosa
**III**	Mild	5–25%	Epiglottic ligament visible
**IV**	Moderate	25–50%	Epiglottic ligament covered
**V**	Severe	>50%	Filled to epiglottic rim
**Pyriform sinus**			
**I**	None	0%	No residue
**II**	Trace	1–5%	Trace coating of the mucosa
**III**	Mild	5–25%	Up wall to quarter full
**IV**	Moderate	25–50%	Up wall to half full
**V**	Severe	>50%	Filled to aryepiglottic fold

**Table 2 diagnostics-12-01897-t002:** Characteristics of participants.

		All Participants (n = 29)
**Age: mean age ± DS**		30.69 ± 6.05
**Sex (females): n (%)**		23 (79.31)
**Speech and language pathologists: n (%)**		20 (68.96)
**Medical doctors: n (%)**		9 (31.03)
**Years of experience: average ± DS**		4.87 ± 3.84
**N FEES ^1^**	**>100: n (%)**	11 (37.93)
	**50–100: n (%)**	11 (37.93)
	**10–49: n (%)**	7 (24.14)
**Participate ^2^ regularly at FEES: n (%)**		18 (62.07)
**Perform ^3^ FEES regularly: n (%)**		6 (20.68)
**Previous clinical experience with the YPRSRS: n(%)**		22 (75.86)
**Post basic training ^4^: n (%)**		12 (41.37)

^1^ How many FEES the rater has participated in/performed; ^2^ To be present in the room when FEES are being performed; ^3^ The execution of the FEES through the passage of the fiberscope; ^4^ e.g., postgraduate diploma, Masters program, PhD.

**Table 3 diagnostics-12-01897-t003:** Construct validity in frames and videos ratings across all raters (n = 29) for all consistencies (thin liquids IDDSI 0, pureed food IDDSI 4 and solid food IDDSI 7).

	Frames Averaged Cohen’s Kappa ± Se	Videos Averaged Cohen’s Kappa ± Se	*t*-Test	
			*t* _(df)_	*p*
**Valleculae**	0.89 ± 0.15	0.79 ± 0.35	*t* _(28)_ = 3.13	0.004
**Pyriform sinus**	0.87 ± 0.03	0.76 ± 0.16	*t* _(28)_ = 4.13	<0.001

Note: based on 15 frames and 15 videos for each location.

**Table 4 diagnostics-12-01897-t004:** Intra-rater reliability in frames and videos ratings across raters who assessed material twice (n = 15) for all consistencies (thin liquids IDDSI 0, pureed food IDDSI 4 and solid food IDDSI 7).

	Frames Averaged Cohen’s Kappa ± Se	Videos Averaged Cohen’s Kappa ± Se	*t*-Test	
			*t* _(df)_	*p*
**Valleculae**	0.93 ± 0.01	0.87 ± 0.03	*t* _(14)_ = 1.90	0.078
**Pyriform sinus**	0.84 ± 0.03	0.86 ± 0.02	*t* _(14)_ = −0.62	0.548

Note: based on 15 frames and 15 videos for each location.

**Table 5 diagnostics-12-01897-t005:** Inter-rater reliability in frames and videos across all raters (n = 29) for all consistencies (thin liquids IDDSI 0, pureed food IDDSI 4 and solid food IDDSI 7).

	Frames Fleiss Kappa ± Se	Videos Fleiss Kappa ± Se	*t*-Test	
			*t* _(df)_	*p*
**Valleculae**	0.85 ± 0.04	0.70 ± 0.09	*t* _(14)_ = −1.59	0.133
**Pyriform sinus**	0.82 ± 0.05	0.67 ± 0.09	*t* _(14)_ = −2.15	0.049

Note: based on 15 frames and 15 videos for each location.

**Table 6 diagnostics-12-01897-t006:** Construct validity in frames and videos ratings across SLPs group and MDs group.

	SLPs (n = 24) Average Cohen’s Kappa ± Se	MDs (n = 5) Avarage Cohen’s Kappa ± Se	*t*-Test	
			*t* _(df)_	*p*
**Valleculae frames**	0.89 ± 0.02	0.89 ± 0.02	*t* _(27)_ = 0.20	0.841
**Valleculae videos**	0.80 ± 0.04	0.77 ± 0.07	*t* _(27)_ = −0.35	0.727
**Pyriform sinus frames**	0.86 ± 0.02	0.90 ± 0.01	*t* _(27)_ = −0.87	0.394
**Pyriform sinus videos**	0.75 ± 0.03	0.82 ± 0.05	t _(27)_ = −0.87	0.392

Note: based on 15 frames and 15 videos for each location.

**Table 7 diagnostics-12-01897-t007:** Intra-rater in frames and videos rating across SLPs group and MDs group.

	SLPs (n = 11) Average Cohen’s Kappa ± Se	MDs (n = 4) Avarage Cohen’s Kappa ± Se	*t*-Test	
			*t* _(df)_	*p*
**Valleculae frames**	0.94 ±0.01	0.92 ± 0.02	*t* _(13)_ = 0.68	0.509
**Valleculae videos**	0.87 ± 0.04	0.89 ± 0.04	*t* _(13)_ = −0.25	0.804
**Pyriform sinus frames**	0.87 ± 0.03	0.74 ± 0.05	*t* _(13)_ = 2.43	0.030
**Pyriform sinus videos**	0.87 ± 0.03	0.83 ± 0.04	*t* _(13)_ = 0.719	0.485

Note: based on 15 frames and 15 videos for each location.

**Table 8 diagnostics-12-01897-t008:** Inter-rater in frames and videos rating across SLPs group and MDs group.

	SLPs (n = 24) Fleiss Kappa ± Se	MDs (n = 5) Fleiss Kappa ± Se	*t*-Test	
			*t* _(df)_	*p*
**Valleculae frames**	0.85 ± 0.04	0.85 ± 0.08	*t* _(27.1)_ = 0.08	0.936
**Valleculae videos**	0.71 ± 0.09	0.65 ± 0.13	*t* _(24.7)_ = 0.40	0.694
**Pyriform sinus frames**	0.81 ± 0.05	0.85 ± 0.06	*t* _(27.5)_ = 0.45	0.657
**Pyriform sinus videos**	0.66 ± 0.09	0.79 ± 0.10	*t* _(27.6)_ = 0.95	0.351

Note: based on 15 frames and 15 videos for each location.

**Table 9 diagnostics-12-01897-t009:** Influence of bolus consistency on construct validity in frames’ and videos’ ratings across all the raters (n = 29).

	Thin Liquids	Pureed Food	Solid Food	*p*-Value		
	Averaged Cohen’s Kappa ± Se	Averaged Cohen’s Kappa ± Se	Averaged Cohen’s Kappa ± Se	Thin Liquids vs. Pureed Food	Thin Liquids vs. Solid Food	Pureed Food vs. Solid Food
**Frames**	0.56 ± 0.04	0.88 ± 0.01	0.85 ± 0.02	<0.001	<0.001	0.711
**Videos**	0.44 ± 0.05	0.88 ±0.02	0.57 ± 0.07	<0.001	0.148	<0.001

Note: pairwise comparison adjusted with Tukey HSD method. Thin liquids viscosity: < 50 mPa·s at 50 s^−1^ and 300 s^−1^. Pureed food viscosity: 2583.3 mPa·s at 50 s^−1^, 697.87 mPa·s at 300 s^−1^.

**Table 10 diagnostics-12-01897-t010:** Influence of bolus consistency on intra-rater reliability in frames and videos ratings across raters who assessed material twice (n = 15).

	Thin Liquids	Pureed Food	Solid Food	*p*-Value		
	Averaged Cohen’s Kappa ± Se	Averaged Cohen’s Kappa ± Se	Averaged Cohen’s Kappa ± Se	Thin Liquids vs. Pureed Food	Thin Liquids vs. Solid Food	Pureed Food vs. Solid Food
**Frames**	0.53 ± 0.06	0.82 ± 0.05	0.89 ± 0.02	<0.001	<0.001	0.518
**Videos**	0.46 ±0.08	0.89 ± 0.03	0.81 ± 0.05	<0.001	<0.001	0.618

Note: pairwise comparison adjusted with Tukey HSD method. Thin liquids viscosity: < 50 mPa·s at 50 s^−1^ and 300 s^−1^. Pureed food viscosity: 2583.3 mPa·s at 50 s^−1^, 697.87 mPa·s at 300 s^−1^.

**Table 11 diagnostics-12-01897-t011:** Influence of bolus consistency on inter-rater reliability in frames and video ratings across all raters (n = 29) at the first assessment for all consistencies.

	Thin Liquids	Pureed Food	Solid Food	*p*-Value		
	Fleiss Kappa ± Se	Fleiss Kappa ± Se	Fleiss Kappa ± Se	Thin Liquids vs. Pureed Food	Thin Liquids vs. Solid Food	Pureed Food vs. Solid Food
**Frames**	0.38 ± 0.09	0.84 ± 0.08	0.82 ± 0.06	0.001	0.001	0.991
**Videos**	0.22 ± 0.09	0.81 ± 0.07	0.48 ± 0.10	<0.001	0.172	0.038

Note: pairwise comparison adjusted with Tukey HSD method. Thin liquids viscosity: < 50 mPa·s at 50 s^−1^ and 300 s^−1^. Pureed food viscosity: 2583.3 mPa·s at 50 s^−1^, 697.87 mPa·s at 300 s^−1^.

## Data Availability

Data supporting reported results can be found at https://doi.org/10.17632/wpzv7w8nrb.1 (accessed on 13 July 2022).

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
