# Peer review of "Reliability and Construct Validity of the Yale Pharyngeal Residue Severity Rating Scale: Performance on Videos and Effect of Bolus Consistency"

_diagnostics, 2022, doi:10.3390/diagnostics12081897_

Round 1
Reviewer 1 Report
Comments to the author The aim of this manuscript is of great importance for healthcare professionals when performing swallowing evaluations. I also agree that these results can encourage improving clinicians’ skills but more important, this manuscript is the first step to stablish a standardized protocol to determine the pharyngeal residue by YPRSRS. There is a need to perform science behind each evaluation to guarantee a quality control not only for professionals but also for the benefit of the patients. The manuscript is well explained and organized. However, I would suggest minor suggestions to be included, which can be of great interest prior its publication. ABSTRACT ï‚· I would recommend to include the viscosity of the products used in the abstract section in mPa·s instead by qualitative measurements (IDDSI). INTRODUCTION ï‚· As the impact of bolus consistency is an aim of the study, I would recommend to the authors to add an explanation on the importance of texture and viscosity modification for patients with swallowing impairments. In addition, I would suggest to clarify the term consistency: IDDSI classification seems to evaluate consistency in a qualitative manner but authors have also reported viscosity in the International System (IS) of units (Pa·s) which is a scientific and objective measurement. MATERIAL AND METHODS ï‚· I would suggest to add the Ethics Committee name and code which has approved the study ï‚· Equipment used to analyze viscosity for each alimentary product selected should be stated. Frames and videos selections ï‚· Page 5 Line 88-92. Viscosity used for this study has been reported at the shear rates of 50s-1 and 300s-1 , which are optimal to observe how the swallowing impact at the oral and pharyngeal cavity, respectively. I would suggest to incorporate an explanation on the selection of these two shear rates (Brito-de la Fuente 2017, Bolívar-Prados 2021). ï‚· Page 5 Line 88-92. I would suggest including the proper alimentary product for each bolus consistency and the brand as authors are trying to validate this specific method. RESULTS Raters characteristics ï‚· Have you observed rating differences according to the raters’ background (SLP, Otolaryngologists and resident medical doctors)? Influence of bolus consistency ï‚· IDDSI classification uses qualitative descriptors and does not use any objective metric nor the IS of units. The International Committee of Medical Journal Editors (ICMJE) recommends reporting this information in both local and International System of Units. I would strongly recommend reporting the results on the tables for viscosity rather than with IDDSI descriptors. Viscosity values allow the reproducibility of the test in any other center in contrast to the IDDSI classification, which each level of consistency average a widely extensive viscosity frame (Bolívar-Prados, 2021). DISCUSSION ï‚· Page 8 Line 275-279. Authors mention that thin liquids (<50mPa·s) presented lower reliability than other consistencies caused by the lower bolus cohesiveness. However, cohesiveness have not been assessed. I would recommend to avoid this comment as cohesiveness can vary according to different factors such as composition and not only to viscosity.

Author Response
Thank you very much for your review of our manuscript and for the valuable comments and suggestions. We have provided our responses (shown in Italics) to your comments in this letter and revised the manuscript accordingly. In the manuscript, review changes are indicated in red. We hope that you will find that the manuscript has been improved.
Reviewer #1
The aim of this manuscript is of great importance for healthcare professionals when performing swallowing evaluations. I also agree that these results can encourage improving clinicians’ skills but more important, this manuscript is the first step to stablish a standardized protocol to determine the pharyngeal residue by YPRSRS. There is a need to perform science behind each evaluation to guarantee a quality control not only for professionals but also for the benefit of the patients.
The manuscript is well explained and organized. However, I would suggest minor suggestions to be included, which can be of great interest prior its publication.
ABSTRACT
- I would recommend to include the viscosity of the products used in the abstract section in mPa·s instead by qualitative measurements (IDDSI).
Following the reviewer's suggestion, in the abstract, the IDDSI terminology was replaced with the viscosity in mPa-s of the products used (page 1 lines 14-15).
INTRODUCTION
- As the impact of bolus consistency is an aim of the study, I would recommend to the authors to add an explanation on the importance of texture and viscosity modification for patients with swallowing impairments. In addition, I would suggest to clarify the term consistency: IDDSI classification seems to evaluate consistency in a qualitative manner but authors have also reported viscosity in the International System (IS) of units (Pa·s) which is a scientific and objective measurement.
We have expanded the introduction on the influence of bolus consistency on swallowing safety and efficacy and included an explanation of the differences between the IDDSI classification and the objective assessment of viscosity expressed in IS units (page 2, lines 76-84).
MATERIAL AND METHODS
- I would suggest to add the Ethics Committee name and code which has approved the study.
According to the reviewer’s comment the information regarding the ethics committee (Ethics Committee of the University of Milan, protocol code 102/02, date of approval 17/11/2020) was added in the methods on page 2, lines 97-98.
- Equipment used to analyze viscosity for each alimentary product selected should be stated.
Information on the equipment to analyze viscosity were added to page 3, lines 124-127. In particular, viscosity analyses were performed with the Haake Viscotester 550 (Thermo Electron GmbH, Dieselstr, Germany); viscosities below 300 mPa·s were performed with the system MV1 (gap:0.96 mm) and viscosities over 300 mPa·s with the system SV1 (gap: 1.45mm).
Frames and videos selections
- Page 5 Line 88-92. Viscosity used for this study has been reported at the shear rates of 50s-1 and 300s-1, which are optimal to observe how the swallowing impact at the oral and pharyngeal cavity, respectively. I would suggest to incorporate an explanation on the selection of these two shear rates (Brito-de la Fuente 2017, Bolívar-Prados 2021).
We thank the reviewer for the suggestion. Required explanations with references have been added to page 3 lines 127-130.
- Page 5 Line 88-92. I would suggest including the proper alimentary product for each bolus consistency and the brand as authors are trying to validate this specific method.
The following products were used: for thin liquids blue-dyed room temperature water, for pureed food Creme Line Vainilla Nutrisens (Nutrisens Italia SRL, Turin, Italy), for solid food half 8g Frollini Monviso (Monviso group SRL, Andezeno(TO), Italy) biscuits. Details about the foods used have been included on page 5, lines 120-124.
RESULTS
Raters characteristics
- Have you observed rating differences according to the raters’ background (SLP, Otolaryngologists and resident medical doctors)?
Following the reviewer's comment, we performed the validity and reliability analyses considering the different backgrounds of the raters. The paragraph and tables with the results were added to the results (page 6, lines 253-263; Tables 6, 7 and 8). No differences were found between SLPs and medical doctors in the Kappa values, except for the intra-rater reliability for FEES frames in the pyriform sinus that was higher for SLPs compared to medical doctors. These results seem to suggest that the YPRSRS can be reliably used by both SLPs and medical doctors working in the field of dysphagia. Results were discussed on page 9, lines 532-546. However, as the comparison based on the raters’ background was not specifically an aim of the study, the sample size of the two groups were unbalanced (n=24 for SLPs and n=5 for medical doctors), results of this study are only preliminary and should be confirmed by further studies.
Influence of bolus consistency
- IDDSI classification uses qualitative descriptors and does not use any objective metric nor the IS of units. The International Committee of Medical Journal Editors (ICMJE) recommends reporting this information in both local and International System of Units. I would strongly recommend reporting the results on the tables for viscosity rather than with IDDSI descriptors. Viscosity values allow the reproducibility of the test in any other center in contrast to the IDDSI classification, which each level of consistency average a widely extensive viscosity frame (Bolívar-Prados, 2021).
Viscosity values in SI units have been added to the tables 9, 10 and 11.
DISCUSSION
- Page 8 Line 275-279. Authors mention that thin liquids (<50mPa·s) presented lower reliability than other consistencies caused by the lower bolus cohesiveness. However, cohesiveness have not been assessed. I would recommend to avoid this comment as cohesiveness can vary according to different factors such as composition and not only to viscosity.
According to the reviewer’s suggestion, the sentence was revised and the words “not being very cohesive” have been removed.
Reviewer 2 Report
This study identifies issues that must be kept in mind when using the Yale Pharyngeal Residue Severity Rating Scale (YPRSRS) in a clinical setting. It mentions that inter-rater reliability is lower when this scale is used with video images than with frames and that some types of bolus make it difficult to use this scale. These are useful information for medical professionals who use this scale to make diagnoses. Thus, the medical value of this paper can be understood. I have pointed out a few minor parts that could be improved, so please consider them.
Line 63
The comparison between frame and video is abruptly presented. A preface is needed to help the reader understand the need to investigate this.
Table 1
What do the numbers in the table mean? Please provide the units needed to understand them.
Author Response
Thank you very much for your review of our manuscript and for the valuable comments and suggestions. We have provided our responses (shown in Italics) to your comments in this letter and revised the manuscript accordingly. In the manuscript, review changes are indicated in red. We hope that you will find that the manuscript has been improved.
Reviewer #2
This study identifies issues that must be kept in mind when using the Yale Pharyngeal Residue Severity Rating Scale (YPRSRS) in a clinical setting. It mentions that inter-rater reliability is lower when this scale is used with video images than with frames and that some types of bolus make it difficult to use this scale. These are useful information for medical professionals who use this scale to make diagnoses. Thus, the medical value of this paper can be understood. I have pointed out a few minor parts that could be improved, so please consider them.
Line 63
The comparison between frame and video is abruptly presented. A preface is needed to help the reader understand the need to investigate this.
We thank the reviewer for the suggestion to improve our manuscript. A preface was added on page 2, line 71-74 to underly the difference between assessment of residues in frames and videos. In clinical practice, assessment is often done in real-time or based on recorded FEES videos. The video assessment, differently to the frames assessment, requires the clinician not only to rate residues’ severity, but also to identify the time to assess residues.
Table 1
What do the numbers in the table mean? Please provide the units needed to understand them.
The numbers represent the percentage of the space of the valleculae or pyriform sinuses filled by the residue. The unit of measurement was added to Table 1.